# Pulmonary Aspergillosis in Humboldt Penguins—Susceptibility Patterns and Molecular Epidemiology of Clinical and Environmental *Aspergillus fumigatus* Isolates from a Belgian Zoo, 2017–2022

**DOI:** 10.3390/antibiotics12030584

**Published:** 2023-03-15

**Authors:** Hanne Debergh, Pierre Becker, Francis Vercammen, Katrien Lagrou, Roel Haesendonck, Claude Saegerman, Ann Packeu

**Affiliations:** 1Mycology and Aerobiology, Sciensano, 1050 Brussels, Belgiumann.packeu@sciensano.be (A.P.); 2Fundamental and Applied Research for Animal and Health (FARAH) Center, ULiège, 4000 Liège, Belgium; 3BCCM/IHEM, Mycology and Aerobiology, Sciensano, 1050 Brussels, Belgium; 4Veterinarian, Centre for Research and Conservation, Antwerp Zoo Society, 2018 Antwerp, Belgium; 5Department of Microbiology, Immunology and Transplantation, KU Leuven, 3000 Leuven, Belgium; 6Department of Laboratory Medicine and National Reference Center for Mycosis, University Hospitals Leuven, 3000 Leuven, Belgium; 7Zoolyx Veterinary Laboratory, 9320 Aalst, Belgium

**Keywords:** *Aspergillus fumigatus*, avian aspergillosis, *Spheniscus humboldti*, antifungal susceptibility testing, MIC, azole resistance, *cyp51A*, microsatellite typing, genotyping, One Health

## Abstract

*Aspergillus fumigatus* is the main causative agent of avian aspergillosis and results in significant health problems in birds, especially those living in captivity. The fungal contamination by *A. fumigatus* in the environment of Humboldt penguins (*Spheniscus humboldti*), located in a Belgian zoo, was assessed through the analysis of air, water, sand and nest samples during four non-consecutive days in 2021–2022. From these samples, potential azole-resistant *A. fumigatus* (ARAF) isolates were detected using a selective culture medium. A total of 28 veterinary isolates obtained after necropsy of Humboldt penguins and other avian species from the zoo were also included. All veterinary and suspected ARAF isolates from the environment were characterized for their azole-resistance profile by broth microdilution. Isolates displaying phenotypic resistance against at least one medical azole were systematically screened for mutations in the *cyp51A* gene. A total of 14 (13.6%) ARAF isolates were identified from the environment (*n* = 8) and from Humboldt penguins (*n* = 6). The TR34/L98H mutation was observed in all resistant environmental strains, and in two resistant veterinary strains. To the best of our knowledge, this is the first description of this mutation in *A. fumigatus* isolates from Humboldt penguins. During the period 2017–2022, pulmonary aspergillosis was confirmed in 51 necropsied penguins, which reflects a death rate due to aspergillosis of 68.0%, mostly affecting adults. Microsatellite polymorphism analysis revealed a high level of diversity among environmental and veterinary *A. fumigatus* isolates. However, a cluster was observed between one veterinary isolate and six environmental strains, all resistant to medical azoles. In conclusion, the environment of the Humboldt penguins is a potential contamination source of ARAF, making their management even more complex.

## 1. Introduction

The saprophytic fungus *Aspergillus fumigatus* is responsible for opportunistic infections affecting birds and mammals, including humans [1]. It can affect a wide variety of species such as domestic, free-ranging or captive wild animals [2]. Aspergillosis is the most common fungal infectious disease affecting penguins in zoos, with up to 99% of cases attributed to *Aspergillus* section *Fumigati*. It represents a major limiting factor for the rehabilitation of penguins in captivity [1,3,4,5,6]. On the contrary, aspergillosis in free-ranging birds is only rarely described [7,8].

The most common route of infection is by inhalation of conidia present in the environment [2,9]. Due to their small size, the spores can bypass the mucociliary clearance by the upper airways [2,9]. As such, they disseminate first in the posterior air sacs, where ideal temperature and oxygen availability conditions are present to allow their germination and the production of hyphae [2]. The important susceptibility of birds, and especially penguins, to aspergillosis also results from the scarcity of immune surveillance cells in the air sac system, and the lack of an epiglottis or diaphragm to block the inhalation of the spores [10,11]. In addition to these predisposing factors, external factors including thermal discomfort, overcrowding or stress also contribute to the high incidence of avian aspergillosis in birds in captivity [2,7,12,13,14], as compared to free-ranging birds [15]. The exposure to high concentrations of *Aspergillus* conidia is presumed to further enhance the occurrence of aspergillosis. High loads of conidia are considered seasonal and influenced by climatic parameters such as temperature, humidity and wind speed [2,16,17]. Furthermore, avian aspergillosis can be acute or chronic, with the former primarily related to young birds and the latter mostly associated with adult animals displaying various levels of immunosuppression, due to external stressors or poor husbandry [10].

Avian aspergillosis can act as a primary infection, affecting mainly the respiratory system. However, infections of the eyes, liver, kidneys, heart, joints and bones have also been described [6,18]. Symptoms associated with aspergillosis in birds are non-specific and include lethargy, weight loss and anorexia, open-beak breathing and dyspnea, coughing, altered vocalization and self-isolation. Moreover, diagnostic tools are not highly reliable and ante mortem diagnosis is thus difficult to make [2,19,20,21,22]. Therefore, a confirmed diagnosis is generally obtained post mortem [23,24]. Treatments often have a poor outcome due to the late diagnosis and the resulting advanced stage of the disease [6]. However, if the disease is suspected, treatment can be initiated with oral administration of itraconazole (ITC) [25]. Prophylaxis with ITC is usually applied for susceptible individuals, e.g., following transfers or antibiotic treatments [2,25]. The One Health concept aims to balance and optimize the health of people, animals and the environment by unifying the topics [26]. Increasing rates of azole resistance in *A. fumigatus* in human medicine are observed [27]. Considering the One Health concept, animals can also be infected by resistant strains. Antifungal treatments in veterinary medicine should be used with care and monitoring of antifungal resistance profiles of veterinary *A. fumigatus* isolates should be adopted [9,28].

*A. fumigatus* is considered as an emerging health threat and was placed in the fungal priority pathogens list by the World Health Organization (WHO) [29], highlighting the importance of its monitoring in human medicine, but also in veterinary health. In a recent five-year study from a tertiary care center in Belgium, an overall prevalence of 7.1% azole-resistance in *A. fumigatus* was detected [30]. Acquired azole resistance is mostly associated with mutations in the *cyp51A* gene, encoding for lanosterol-C14-α-demethylase [27,31]. Other resistance mechanisms exist, such as mutations in *hmg1*, *cdr1B*, *HapE* or overexpression of efflux pumps [32,33,34,35,36]. The intensive use of agricultural azole antifungals has been linked to the development of azole resistance against medical antifungals [31,37]. The surveillance of azole resistance in *A. fumigatus* in clinical strains is a common practice, however, this is not the case for avian aspergillosis with only few epidemiological studies published [6,12,13,38]. In this respect, the study of the genetic diversity in *A. fumigatus* is a valuable tool to better understand the transmission routes involved. This study aimed to assess the impact of environmental *A. fumigatus* contamination on the clinical incidence of avian aspergillosis in Humboldt penguins in a Belgian zoo. The susceptibility patterns towards medical azoles and the mutations in the *cyp51A* gene were studied, alongside the genotyping of the strains, in order to estimate the epidemiology of the infections.

## 2. Results

### 2.1. Environmental Aspergillus fumigatus Sampling inside the Penguin Enclosure

Cultures obtained on a malt extract + chloramphenicol (MC) medium from environmental air sampling at four predefined locations(Figure 1) revealed a mean *A. fumigatus* burden in spring and summer of 41.5 and 56.5 CFU/1000 L, respectively, and 15.75 and 22.75 CFU/1000 L in autumn and winter, respectively. These differences were, however, not significant and the overall *A. fumigatus* contamination (log_10_ CFU/1000 L) statistically similar between the different study time points for all four study locations (*p*-value = 0.27) (Figure 2). All *A. fumigatus* colonies isolated from a malt extract + chloramphenicol + 4 mg/L tebuconazole (MC + T) medium (*n* = 68) were stored and further analyzed in this study. All isolates were confirmed as *A. fumigatus* using MALDI-TOF MS with a score ≥ 2.0. The samples from sand, water and nest swabs were negative for *A. fumigatus*.

### 2.2. Clinical Incidence of Avian aspergillosis in Humboldt Penguins in a Belgian Zoo

Since the start of the Humboldt penguin program in 2013, a total of 214 animals were counted, of which 189 have died (88.3%) until December 2022. Clinical samples from 2017–2022 were included in this study. During this 6-year period, 96 Humboldt penguins died and necropsy was performed on 75 carcasses (78.1%) (Table 1). Necropsy was not performed on the remaining animals due to alternative causes of death (rotten in nests at few days old or trampled). In 2018, the highest death rate was observed, with 37 dead animals (37/78; 47.4%) (Table 1). A total of 34 were necropsied; and in 26 cases, pulmonary aspergillosis was confirmed as the cause of death (26/34; 76.5%). Over the course of 6 years, pulmonary aspergillosis was confirmed in 51 necropsied penguins, which reflects a death rate due to aspergillosis of 68.0% (51/75). Five of the penguins included in this study received ITC at 20 mg/kg per day as treatment for an extended period of time (Table 2). The amount of breeding couples in 2017 and 2018 was not known. In 2019 and 2022, no breeding program was initiated, whereas in 2020 and 2021, respectively, four and eight breeding couples were present. 

*A. fumigatus* isolates (*n* = 35) were obtained from Humboldt penguins (*n* = 29), South African penguin (*Spheniscus demersus*) (*n* = 1), Red-billed blue magpie (*Urocissa erythrorhyncha*) (*n* = 1), Chilean flamingo (*Phoenicopterus chilensis*) (*n* = 1), Chestnut-backed thrush (*Geokichlia dohertyi*) (*n* = 1), Crested oropendola (*Psarocolius decumanus*) (*n* = 1) and Crested partridge (*Rollulus rouloul*) (*n* = 1) that died between 2017 and 2022. A total of 66% of those isolates were obtained from 2018 (23/35). Other isolates were obtained in 2017 (*n* = 2), 2019 (*n* = 1), 2020 (*n* = 2), 2021 (*n* = 6) and 2022 (*n* = 1).

### 2.3. Broth Microdilution Antifungal Susceptibility Testing and cyp51A Sequencing

All clinical (*n* = 35) and environmental (*n* = 68) *A. fumigatus* isolates were subjected to broth microdilution antifungal susceptibility testing. A total of six clinical (17.14%) and eight environmental (11.76%) *A. fumigatus* isolates displayed resistance against at least one medical azole (Table 3).

All *A. fumigatus* isolates showing resistance against at least one medical azole were further characterized by *cyp51A* sequencing. Two clinical isolates showed the presence of the TR34/L98H mutation (Table 3) and one displayed several nucleotide mutations, resulting in three amino acid substitutions F46Y, M172V, E427K. Three other clinical isolates did not show any mutations known to cause resistance in the *cyp51A* gene. All environmental isolates showed the presence of the TR34/L98H mutation and one (21-0503) showed an additional G54R nucleotide mutation.

### 2.4. Microsatellite Genotyping of the Aspergillus fumigatus Isolates

Genotyping was performed on a selection of 45 *A. fumigatus* isolates (Figure 3, Table A1). The discriminatory power of the combined markers reached 98.8. A total of 21 veterinary isolates were included—18 from Humboldt penguins and 3 from other avian species (*Psarocolius decumanus*, *Spheniscus demersus* and *Rollulus rouloul*) to investigate if transmission between bird species was possible. No identical genotype was shared between the veterinary isolates. Twenty-four environmental isolates, representing both susceptible and resistant strains, were also analyzed and resulted in 20 different genotypes. The same genotype was found in four environmental strains (21-0515, 21-0516, 21-0517 and 21-0518), and were closely related to two other genotypes: a first one shared by two environmental strains (21-0503 and 21-0510), and a second one corresponding to an isolate (21-0524) originating from a Humboldt penguin (Figure 3, Table A1). Genotypes within this cluster differed by only one marker (STRAf2A). All seven isolates in this cluster were resistant against at least one medical azole (Figure 3). The six environmental strains originated from the study time point in autumn (October 2021). The Humboldt penguin (21-0524) died during the same period, in August 2021 (Table 2).

## 3. Discussion

Aspergillosis in captive birds plays an important role which can pose problems in their management. All avian aspergillosis cases in our study arose by *A. fumigatus*, confirming its role as the main causative fungal agent of avian aspergillosis in penguins, as previously described [12,40]. The naturally elevated body temperature of the birds, ranging from 39 to 41 °C, combined with the thermotolerance of *A. fumigatus*, favor its incidence. The second most common agent of bronchopulmonary aspergillosis in birds is *A. flavus,* accounting for about 5% of avian aspergillosis cases [11], but this species was not detected in this study.

In the present study, the mortality due to aspergillosis reached 86.7% in 2017 and 76.5% in 2018. In 2018, we see a clear rise in aspergillosis incidence compared to the other years). This might be explained by the extreme weather conditions in 2018, especially due to the high temperatures in summer: 65% of the deaths in 2018 occurred during the summer months. Additionally, the population size at that time was large and a breeding program was operative in 2018. The effect of overcrowding was demonstrated in a study performed over 6 years in Magellanic penguins (*Spheniscus magellanicus*), where 65% of aspergillosis cases took place during the year with the highest population density [5]. Measures were taken in the Belgian zoo in 2019 to lower the mortality of aspergillosis: ventilation holes were installed in the nests, monitoring of the water temperature below 20 °C to mimic their natural habitat, shade was provided in summer, breeding program was interrupted and direct contacts between visitors and penguins were discontinued. The breeding program started again in 2020 with 4 breeding couples in 2020 and 8 in 2021. The mortality in penguins increased again when breeding restarted. Over the period 2017–2021, it appeared that the aspergillosis incidence was 3-fold higher in animals rearing young (*n* = 22, 12 female and 10 male) than those that did not (*n* = 7, 5 female and 2 male). The highest mortality was observed in adults, which is in contrast with previous findings, where most of the cases of aspergillosis occur in juveniles [5,13,41]. Avian aspergillosis in adult penguins is mostly due to the chronical form and is linked to immune suppression [6]. Noteworthy, juveniles younger than 2 months (*n* = 20) were not tested in this study for the presence of aspergillosis.

Environmental sampling in the penguin housing did not show significant differences between the four study time points. However, mean fungal loads in the environment were slightly more elevated in spring and summer. This is in line with the findings of Cateau et al. reporting higher fungal burden in September compared to April and December [13]. Similarly, a study performed on African Penguins in the Maryland Zoo in Baltimore, described higher environmental fungal load during the warmer period from the end of spring to the beginning of autumn [42].

Artificial nests made up of plastic in the synthetic rock formations did not reveal the presence of *A. fumigatus* in this study. Here, only two nests were consistently sampled on 3 out of the 4 sampling days, which might explain the absence of *A. fumigatus*. However, unpublished data from the zoo obtained on a larger sampling campaign of all the nests (*n* = 25), confirmed that in the period of 2017–2018, *A. fumigatus* colonies were found in all nests, with higher average counts from May to September, as compared to October to March. This is in agreement to the study of Cateau et al. which found a high fungal load in the nests [13]. 

Azole resistance in *A. fumigatus* represents an emerging problem in human and veterinary medicine and was detected in 14 isolates, both from the environment and from penguins. Most of the mutations conferring azole resistance in *A. fumigatus* are found in the *cyp51A* gene encoding for lanosterol-C14-α-demethylase, the target protein of azole drugs [27]. *Cyp51A* mutations can be tandem repeats (TR) in the promotor region of the gene, single-nucleotide polymorphisms (SNPs), or both [27]. Many articles describe the TR34/L98H mutation conferring resistance to azole drugs, which has been linked to the intense use of agricultural azoles for crop protection [31]. The zoo is located near a larger city, however, it is also surrounded by many agriculturally cultivated plots. In this study, the TR34/L98H mutation was found in both clinical and environmental samples. This mutation usually leads to pan-azole resistance phenotype in human clinical samples, which can be seen in several samples in this study (Table 3). In contrast, the most common mutations in *cyp51A* leading to azole-resistance that develop during antifungal treatment, occur in amino acid sites G54, G138, M220, and G448 [43]. Prophylaxis with azoles in distressed penguins, as well as the treatment of aspergillosis in captive penguins, are very common [25]. Several penguins included in our study received treatment for extended periods due to an increased chance of disease development.

However, only the TR34/L98H mutation was detected, in two strains isolated from Humboldt penguins, suggesting that the resistance of these strains was acquired from the environment. The isolate 21-680 harbored several other amino acid substitutions in the *cyp51A* gene: F46Y, M172V, E427K. The combination of these amino acid substitutions was reported in approximatively 10% of all *A. fumigatus* isolates tested worldwide [44], including in patients receiving azole treatment [45]. Generally, they display elevated MIC values for the medical azoles compared to the wild-type (WT) *cyp51A* [45]. However, their susceptibility profiles are inconsistent and were described as both azole-susceptible or resistant by different authors [31,44,45,46,47,48,49,50,51]. No known resistance conferring mutations in the *cyp51A* gene were found in the remaining veterinary isolates. Other mechanisms, such as mutations in other genes [32,36,40] or efflux pumps [34] could be involved in the decreased susceptibility of these isolates to POSA. All eight environmental azole-resistant strains harbored the TR34/L98H mutation. Three of them displayed the typical pan-azole phenotype, whereas the remainder were resistant against ISA and POSA, but had MIC values in the area of technical uncertainty (ATU) for VOR and ITC.

Genotyping of *A. fumigatus* isolates in this study revealed a broad diversity in both environmental and veterinary strains, suggesting independent events of contamination. Additionally, the azole-resistant strains were not all closely related to each other, indicating that the resistance was acquired multiple times and has different origins. No relation was observed between the veterinary isolates included in this study. In contrast, Cateau et al. found identical genotypes among veterinary isolates, but their sampling was performed within a short timespan [13]. A veterinary strain from a Humboldt penguin (which died in August 2021), however, clustered with six environmental isolates from the study time point in autumn. Moreover, all seven isolates of this cluster were resistant against at least one medical azole and harbored the TR34/L98H mutation. This suggests that the Humboldt penguin acquired the strain from the environment. Interestingly, isolates 21-0503 and 21-0510 had identical genotypes, but differed in MIC values and *cyp51A* sequencing. Isolate 21-0503 indeed had an additional G54R amino acid mutation, alongside the TR34/L98H mutation. The observed MIC values were higher for 21-0510 than for 21-0503 for ISA and POSA. Additionally, two strains (21-0428 and 21-0488) originating from the same animal had different unrelated genotypes, indicating that Humboldt penguins can be infected by multiple *A. fumigatus* strains. Altogether, within this study we were able to capture a small proportion of the large diversity present in the environment and veterinary *A. fumigatus* strains present in the Belgian zoo.

This research has several limitations. The first is the timeframe of environmental sampling which was performed on 4 non-consecutive days. This gives a limited view on the seasonality since single time point measurements depend on many different aspects such as temperature; wind or humidity. Future studies should therefore consider continuous long-term sampling. Secondly, environmental sampling was only performed in 2021 and 2022; while the majority of the veterinary strains were isolated during the previous years.

There is limited evidence in humans that the infection can be spread from patient to patient [52]. There is, however, no evidence reported yet of such events in animals. However, considering the anatomy of the respiratory system of birds, and the high fungal burden in their air sacks/lungs, it could also be true for birds. This would need more research with multiple time point sampling of penguins living in the same habitat. However, this might be a challenge considering the invasive nature of sampling living penguins and the poor reliability of diagnostic tools ante mortem.

The One Health concept envisages a tripartite health system based on the environment, animals and humans and their mutual interactions. This concept directs us towards a more holistic approach in the surveillance of infectious diseases on a global scale. The interactions between the environment, the ubiquitous mold *A. fumigatus*, and the birds, illustrate such complex ecosystem. The latter is changing in an accelerated manner due to global warming, which should be considered when addressing research on infectious diseases. In this paper, we were able to show a probable interaction between the environment and animals, but transmission between animals was not evidenced. However, we observed the same resistance profile and gene mutations in the environment and in the animals, which are also observed in human isolates [31]. The effect of extreme weather conditions was demonstrated. We could predict these phenomena to occur more often with the effects of global warming.

In conclusion, this report contributes to a better understanding of the molecular epidemiology of avian aspergillosis in penguins, dominated by *A. fumigatus*. It is also the first report of a TR34/L98H mutation in *A. fumigatus* isolates obtained from penguins, showing the relevance of monitoring azole resistance of *A. fumigatus* in veterinary science. Genotyping revealed infection by multiple *A. fumigatus* strains in the same penguin individual, as well as a clustering between environmental and veterinary isolates. More frequent sampling could provide more insight on the diversity and possible transmission of *A. fumigatus* between Humboldt penguins and their environment. The high mortality rates due to aspergillosis observed in this study also question the best combination of practices in captive penguins management.

## 4. Materials and Methods

### 4.1. Environmental Aspergillus fumigatus Sampling within the Penguin Enclosure 

Environmental sampling was conducted between April 2021 and January 2022 at a Belgian zoo. The maximal population size of the group was 52 in 2021 and 28 in 2022. Since the start of the Humboldt penguin program in 2013, the zoo welcomed a total of 214 animals. The penguin habitat, which is exclusively outdoors, includes a temperature-monitored swimming pool (<20 °C) and artificial rock formations with 25 built-in nests (Figure 4). The entire habitat is roofed with a steel wired net to prevent the entry of other birds.

Four study time points of environmental sampling were performed on April 1st 2021, 29 June 2021, 13 October 2021 and 12 January 2022, covering spring, summer, autumn and winter, respectively. At each sampling day, four types of samples were taken: air, water, sand and nest surface (except for nest swabs on April 1st since breeding couples were present and could not be disturbed). Sand and air were taken at four predefined locations (Figure 1). Two nest samples and one water sample of the pool were taken (Figure 1).

### 4.2. Isolation of Aspergillus fumigatus from Environmental Samples

All samples were plated on two media: malt extract agar + chloramphenicol (MC) and MC supplemented with 4 mg/L of tebuconazole (MC + T). The plates were incubated at 48 °C ± 1 °C for 48 h ± 2 h to prevent the growth of most environmental fungi. For the air samples, a total of 1000 L of air was impacted on each medium using the MAS-100 NT™ impactor (Merck^®^, Darmstadt, Germany) with a rate of 100 L/min. Quantitative results were expressed as log10 CFU/1000 L air. From each sand sample, 1 g was dissolved in 9 mL of 0.85% NaCl + 0.01% Tween 20 solution. After thorough vortexing, 100 microliters were plated onto the two media. A total of 100 mL of water from the swimming pool was filtered using a nitrocellulose membrane filter (0.45 µm, Sartorius, Göttingen, Duitsland) in two times, after which each filter was placed on a different medium. For the surface sampling of the nests, swabs were taken on the inside of the nests, (floor and walls) and inserted in 1 mL of Amies liquid (eSwab, Copan, Menen, Belgium). Following a 1 min vortexing step, 100µL of each suspension was seeded onto the two media.

The MC medium was used to determine the total number of *A. fumigatus* isolates. Fungal colonies on MC + T were isolated and identified for further analysis to detect potential azole-resistant *A. fumigatus* (ARAF) isolates. The microscopic and macroscopic features of each colony were used to identify every *A. fumigatus* isolate. Matrix-assisted laser desorption ionization-time of flight mass spectrophotometry (MALDI-TOF MS) was used to confirm the identity of the suspected ARAF isolates that grew on MC + T [53].

### 4.3. Clinical Incidence of Avian aspergillosis in Humboldt Penguins

Aspergillosis incidence in the Humboldt penguins in the zoo was assessed for the period 2017–2022. Necropsy was performed on all dead animals older than 2 months with suspected aspergillosis (Figure 5). Confirmed cases of pulmonary aspergillosis were verified by light microscopy using a lactophenol cotton blue stain and culture of *A. fumigatus* of the lungs and/or the surface of the air sacs.

### 4.4. Broth Microdilution Antifungal Susceptibility Testing and cyp51A Sequencing

All strains able to grow on MC + T were tested by broth microdilution method according to the European Committee on Antimicrobial Susceptibility Testing (EUCAST) guidelines [54]. Briefly, a cell suspension of 1–5 × 10^6^ CFUs (colony-forming unit) per ml was prepared in 10 mL of saline water (8.5 g/L NaCl) from a 5 day old subculture on Sabouraud chloramphenicol agar tube. Subsequently, 1 mL of the cell suspension was added and mixed with 10 mL of RPMI-1640 medium (Sigma-Aldrich, Saint-Louis, MO, USA). A total of 100 µL of this cell suspension was added to each well of a 96-well plate containing 100 µL of serial dilutions of the antifungals and a control. The plates were incubated at 35 °C ± 1 °C for 48 h. The minimal inhibitory concentration (MIC) of four medical azoles (itraconazole (ITC), voriconazole (VRC), posaconazole (POSA) and isavuconazole (ISA)) was determined on suspected ARAF strains. The MIC was determined visually as the lowest concentration of antifungal drugs causing complete inhibition of fungal growth. Candida krusei (IHEM 9560 = ATCC 6258), Candida parapsilosis (IHEM 3270 = ATCC 22019) and A. fumigatus (IHEM 28944 = ATCC 204305) were used as quality control strains. Azole resistance was defined according to the EUCAST clinical breakpoints (v10.0) [55], for ITC and VOR with MIC > 1 mg/L, POSA MIC > 0.25 mg/L and ISA MIC > 2 mg/L).

### 4.5. Genotyping

A selection of environmental and clinical strains were analyzed by microsatellite polymorphism genotyping using three multiplex PCRs. A total of nine microsatellite markers consisting of di-, tri-, or tetranucleotide short tandem repeats (STR) were used [56]. Fungal DNA extraction was performed by freeze-drying the cultures and mechanically breaking the cells by bead-beating. DNA was then extracted using the ZR Fungal/Bacterial DNA MiniPrep Kit (Zymo Research) following the manufacturer’s instructions. Genotyping was performed by Genoscreen (Lille, France) using the PCR conditions described by De Valk et al. [56] with the fluorophores 6FAM/HEX/NED. The size of the amplicons was determined with a ABI 3730XL genetic analyzer using the GeneScan 500 ROX size standard (ABI) and the GeneMapper v5.0 software. The size of each microsatellite fragment was measured to determine the number of repetitions for each marker according to de Valk et al. [56]. All results are reported as repeat numbers. The relatedness of the strains was estimated by a minimum spanning tree analysis in Bionumerics 8.0 (Applied Maths, St-Martens-Latem, Belgium). The discriminatory power of the microsatellite markers was calculated using the Simpson index of diversity (Hunter 1990).

### 4.6. Statistics

Statistical analysis was performed to achieve global comparison between each study time point, using the non-parametric Friedman test. The statistical and graphics software R (version 4.2.0) was used. The significance level was set at *p*-value < 0.05.

## Figures and Tables

**Figure 1 antibiotics-12-00584-f001:**
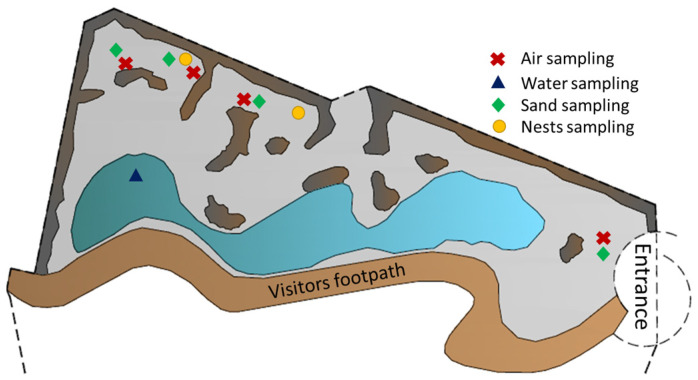
Map of the Humboldt penguin enclosure. Dark brown represents artificial rock formations with 25 built-in nests in total. Light brown represents the visitors pathway. Gray represents the sand. Four sample types were taken: red cross = air sample; blue triangle = water; green diamond = sand; yellow circle = nest swab.

**Figure 2 antibiotics-12-00584-f002:**
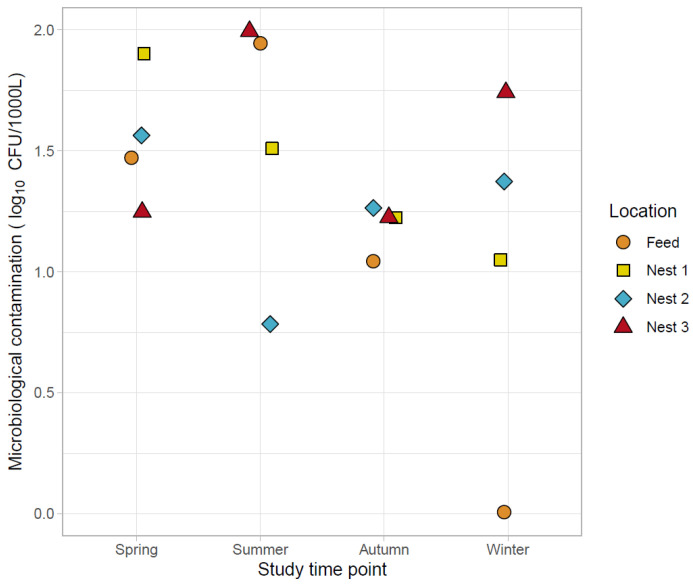
Environmental analysis in the enclosure of the Humboldt penguins. Scatter plot showing the microbiological contamination of *A. fumigatus* in air samples (log10 CFU/1000 L air) grown on a MC medium. Samples were taken during four non-consecutive study days covering all seasons, in four different locations (feeding stage and three locations near the nests).

**Figure 3 antibiotics-12-00584-f003:**
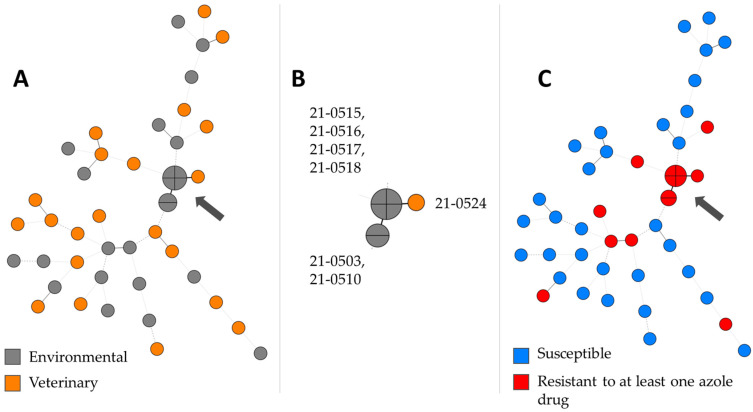
Minimum spanning tree displaying the 41 different genotypes obtained from 45 *Aspergillus fumigatus* isolates, based on 9 variable-number tandem-repeat loci. The genotypes are represented by circles. The length and thickness of the connecting lines between the circles show the similarity between the profiles. (**A**) Mapping of the source of the isolate (i.e., isolated from the environment or from an infected animal). (**B**) Detailed map of the cluster indicating the strains ID. (**C**) Mapping of the resistance profile towards azole drugs (i.e., susceptible to all tested drugs, or resistant to at least one tested drug). The arrow shows the seven clustered isolates.

**Figure 4 antibiotics-12-00584-f004:**
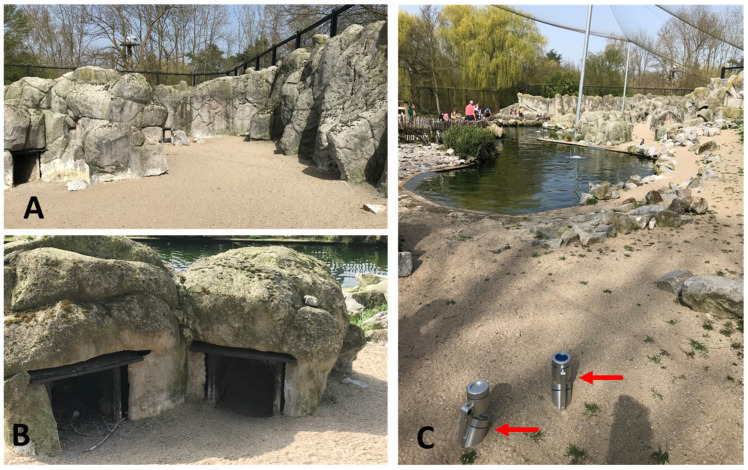
Presentation of the Humboldt penguin enclosure. (**A**) Artificial rocks with built-in nests; (**B**) two nests made up of plastic; (**C**) temperature-controlled pool surrounded by sand and artificial rock formations. Air sampling was performed using MAS-100 NT air samplers (red arrows).

**Figure 5 antibiotics-12-00584-f005:**
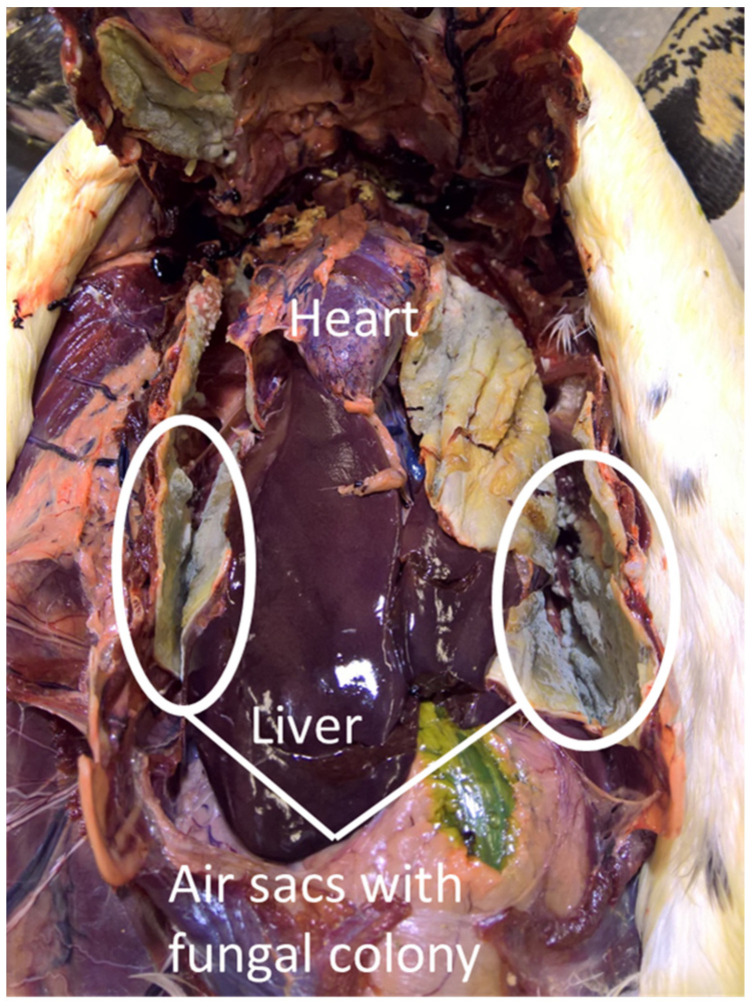
Gross lesions of aspergillosis found in a captive Humboldt penguin. View of the opened coelomic cavity with dissected air sacs that show confluent green to gray velvety fungal growth in the air sac.

**Table 1 antibiotics-12-00584-t001:** Population information of the Humboldt penguins in a Belgian zoo from 2017 to 2022.

Year	Living Animals	Dead Animals	Death < 2 Months Old	Mortality Rate (%)	Necropsy	Confirmed Aspergillosis	Aspergillosis (%) *	Incidence (%) **
2017	99	27	11	27/99 (27.3)	15	13	86.7	13.1
2018	78	37	3	37/78 (47.4)	34	26	76.5	33.3
2019	41	2	0	2/41 (4.9)	2	1	50.0	2.4
2020	45	6	2	6/45 (13.3)	4	1	25.0	2.2
2021	52	20	4	20/52 (38.5)	16	7	43.8	13.5
2022	28	4	0	4/28 (14.3)	4	3	75.0	10.7

* Percentage of aspergillosis was calculated as number of confirmed pulmonary aspergillosis cases per number of necropsies. ** Mortality incidence directly attributable to aspergillosis was calculated as number of confirmed aspergillosis cases per number of living animals.

**Table 2 antibiotics-12-00584-t002:** Demography of the bird specimens included in this study, and minimum inhibitory concentration towards antifungal drugs of the *A. fumigatus* strains isolated from the autopsied bodies.

ID A. Fumigatus Strain	VOR	ITC	ISA	POSA	Host Species	Date of Death	Age	Treatment Since **
21-0659	1	0.5	0.5	0.125	Humboldt penguin	28/12/2017	16y10m5d	N/A
21-0660	1	0.25	1	0.25	Humboldt penguin	29/12/2017	20y7d	N/A
21-0029	0.25	0.25	0.25	0.125	Humboldt penguin	12/11/2018	11y5m18d	N/A
21-0030	0.5	0.25	0,5	0.125	Humboldt penguin	12/12/2018	11y5m18d	N/A
21-0661	1	0.5	1	0.25	Humboldt penguin	1/01/2018	20y9m23d	N/A
21-0662	4 *	>16 *	4 *	0.5 *	Humboldt penguin	26/02/2018	8y9m27d	16/02/2017
21-0663	1	0.5	0.5	0.25	Humboldt penguin	2/03/2018	9m7d	N/A
21-0664	0.5	0.25	0.5	0.25	Humboldt penguin	29/03/2018	19y17d	N/A
21-0665	2	0.5	1	0.25	Crested partridge	4/04/2018	5m15d	N/A
21-0666	0.5	0.25	0.5	0.125	Humboldt penguin	7/04/2018	13y11m23d	N/A
21-0667	1	0.5	0.5	0.25	Humboldt penguin	9/05/2018	14y1m2d	2017 †
21-0668	1	0.5	0.5	0.25	Humboldt penguin	18/07/2018	11y1m17d	N/A
21-0669	0.5	0.25	0.5	0.064	Humboldt penguin	19/07/2018	11y1m24d	N/A
21-0670	1	0.55	1	0.25	Humboldt penguin	25/07/2018	11y2m	N/A
21-0671	0.5	0.25	0.5	0.25	Humboldt penguin	7/08/2018	2y1m6d	N/A
21-0672	0.5	0.5	0.5	0.25	Humboldt penguin	9/08/2018	2y1m8d	N/A
21-0673	0.5	0.5	0.5	0.125	Humboldt penguin	8/08/2018	12y1m24d	N/A
21-0674	0.5	0.5	2	0.25	Humboldt penguin	17/08/2018	7y4m7d	N/A
21-0675	1	0.5	2	0.25	Humboldt penguin	5/09/2018	17y4m15d	N/A
21-0676	1	1	1	0.5 *	Humboldt penguin	5/09/2018	7y3m11d	N/A
21-0677	0.5	0.5	1	0.25	Humboldt penguin	4/09/2018	3y4m13d	N/A
21-0678 *	0.5	0.5	1	0.5 *	Humboldt penguin	13/09/2018	9y5m1d	N/A
21-0679	1	1	2	0.25	Humboldt penguin	22/09/2018	22y5m1d	16/2/2017
21-0680	1	1	1	0.5 *	Humboldt penguin	28/09/2018	17y5m8d	2018 †
21-0034	0.5	0.25	0.5	0.125	Red-billed blue magpie	17/11/2019	4m, 4d	N/A
21-0035	0.25	0.25	0.25	0.125	Chilean flamingo	22/02/2020	45y7m19d	N/A
21-0036	0.5	0.25	0.5	0.125	Chestnut-backed thrush	8/10/2020	6y1m5d	N/A
21-0353	0.5	0.25	0.5	0.125	Crested oropendola	5/03/2021	6y11m21 d	N/A
21-0428/21-0488	0.25	0.25	0.25	0.125	Humboldt penguin	26/05/2021	12y1m17d	2020 †
21-0494	2	1	2	0.5 *	Humboldt penguin	1/08/2021	4y3m3d	N/A
21-0523	0.5	0.25	0.5	0.125	Humboldt penguin	6/09/2021	20y7m23d	N/A
21-0524	4 *	2	4 *	0.5 *	Humboldt penguin	1/08/2021	5y3m12d	N/A
21-0525	0.5	0.5	0.5	0.125	African penguin	12/08/2021	11y3m2d	N/A
22-0592	0.25	0.25	0.5	0.032	Humboldt penguin	29/08/2022	22y4m12d	N/A

N/A = not applicable, VOR = voriconazole, ITC = itraconazole, ISA = isavuconazole, and POSA = posaconazole; * indicates resistance following the EUCAST clinical breakpoints for fungi, v10.0 [39]; ** starting date of treatment with 20 mg/kg itraconazole daily until death; † exact starting date of treatment is unknown.

**Table 3 antibiotics-12-00584-t003:** Antifungal susceptibility testing results of the isolates displaying antifungal resistance against at least one medical azole, and associated *cyp51A* mutation.

Source	ID Strain	Date of Isolation	VOR	ITC	ISA	POSA	*cyp51A* Mutation
Environmental	21-0468	29/06/2021	2 *	2 *	4 *	0.5 *	TR34/L98H
	21-0503	13/10/2021	4 *	>16 *	4 *	0.5 *	TR34/L98H, G54R
	21-0506	13/10/2021	4 *	>16 *	4 *	0.5 *	TR34/L98H
	21-0510	13/10/2021	4 *	>16 *	16 *	2 *	TR34/L98H
	21-0515	13/10/2021	2 *	2 *	4 *	0.5 *	TR34/L98H
	21-0516	13/10/2021	2 *	2 *	4 *	0.5 *	TR34/L98H
	21-0517	13/10/2021	2 *	2 *	4 *	0.5 *	TR34/L98H
	21-0518	13/10/2021	2 *	2 *	4 *	0.5 *	TR34/L98H
Veterinary	21-0662	26/022018	4 *	>16 *	4 *	0.5 *	TR34/L98H
	21-0676	5/09/2018	1	1	1	0.5 *	no known mutations found
	21-0678	13/09/2018	0.5	0.5	1	0.5 *	no known mutations found
	21-0680	28/09/2018	1	1	1	0.5 *	F46Y, M172V, E427K
	21-0494	1/08/2021	2 *	1	2 *	0.5 *	no known mutations found
	21-0524	1/08/2021	4 *	2 *	4 *	0.5 *	TR34/L98H

*** phenotypical resistance according to the EUCAST clinical breakpoints (v10.0) [39].

## Data Availability

All available data are displayed in this paper.

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
