# Peer review of "Pulmonary Aspergillosis in Humboldt Penguins—Susceptibility Patterns and Molecular Epidemiology of Clinical and Environmental Aspergillus fumigatus Isolates from a Belgian Zoo, 2017–2022"

_antibiotics, 2023, doi:10.3390/antibiotics12030584_

Round 1
Reviewer 1 Report
The study on “Aspergillosis in Humboldt Penguins – Susceptibility Patterns and Molecular Epidemiology of Clinical and Environmental Isolates from a Belgian Zoo, 2017-2022” provides deep scientific insights about the azole resistant Aspergillosis in Humboldt Penguins. The article has been well written with scientifically relevant facts.
I would suggest minor corrections:
1. Authors should include the reference “Medical Mycology, Volume 60, Issue Supplement_1, September 2022, myac072P467, https://doi.org/10.1093/mmy/myac072.P467”
2. Italicize A. fumigatus in materials and methods subsections
Some Queries needs to be addressed:
3. The results explained in lines 106-111 are not clear. Here the results first showed different mean CFU for different seasons. However, in the next sentence authors stated that “overall A. fumigatus contamination (log10 CFU/1000L) was not significantly different between the different study time points.
4. In the methodology section, authors have showed the isolation of fungual colonies from sand, water, and surface of nests were also carried out, while no results for these have been included in the article.
5. Is there any explanation for why the microbiological contamination of air samples from 3 different nests are significantly different from each other during summer (figure 3)?
6. How it is relevant to correlate the environmental fungal contamination level which was done for 1 year with the animal fungal load or drug resistance which was carried out over 6 years?
7. Is there any specific reason why only clinical samples from Humboldt Penguins were screened for antifungal susceptibility testing?
Reviewer 2 Report
In the manuscript entitled "Aspergillosis in Humboldt Penguins – Susceptibility Patterns and
Molecular Epidemiology of Clinical and Environmental Isolates from a Belgian
Zoo, 2017-2022”.
The authors have described the aspergillosis in Humboldt Penguins, and also Aspergillus environmental isolates. This manuscript is interesting and can be educational. By the way, there are some points that need revision.
1. The title is unclear! You should mention the kind of aspergillosis in penguins and mention Aspergillus environmental isolates too!
2. Line 21: “A. fumigatus” to be “Aspergillus fumigatus” for the first name.
3. Line 34: Please mention the kind of aspergillosis in penguins.
4. Line 53: this sentence doesn’t have any reference! You can add this related reference too: a. Assessment of indoor and outdoor airborne fungi in an Educational, Research and Treatment Center. Italian Journal of Medicine. 2017; 11: 52-56.
5. Lines 81-84: this sentence is unclear, and needs an English revision.
6. Line 85: “Aspergillus fumigatus” to be “A. fumigatus” in the rest of the text.
7. Table 1: “confirmed aspergillosis” what do you mean? At least mention it in the subtitle. “Aspergillosis” which kind?
8. Line 261: “Most of the mutations conferring azole resistance in …, the target protein of azole drugs.” this sentence doesn’t have any reference! You can add this related reference too: b. High Prevalence of Clinical and Environmental Triazole Resistant Aspergillus fumigatus in Iran: Is It a Challenging Issue? J Med Microbiol. 2016; 65: 468-475.
9. Line 393: Please mention the methods used for microscopic examinations exactly.
Round 2
Reviewer 1 Report
Authors have addressed all comments, the manuscript can now be accepted for publication.